

# Putatively asexual chrysophytes have meiotic genes: evidence from transcriptomic data

Diana Kraus[1], Jingyun Chi[1], Jens Boenigk[2], Daniela Beisser[2,3], Nadine Graupner[2] and Micah Dunthorn[1,3,4]

[1] Department of Ecology, University of Kaiserslautern, Kaiserslautern, Germany
[2] Department of Biodiversity, University of Duisburg-Essen, Essen, Germany
[3] Centre for Water and Environmental Research (ZWU), University of Duisburg-Essen, Essen, Germany
[4] Department of Eukaryotic Microbiology, University of Duisburg-Essen, Essen, Germany

## ABSTRACT

Chrysophytes are a large group of heterotrophic, phototrophic, or even mixotrophic protists that are abundant in aquatic as well as terrestrial environments. Although much is known about chrysophyte biology and ecology, it is unknown if they are sexual or not. Here we use available transcriptomes of 18 isolates of 15 putatively asexual species to inventory the presence of genes used in meiosis. Since we were able to detect a set of nine meiosis-specific and 29 meiosis-related genes shared by the chrysophytes, we conclude that they are secretively sexual and therefore should be investigated further using genome sequencing to uncover any missed genes from the transcriptomes.

# INTRODUCTION

The Chrysophyceae Pascher 1914 are a morphologically diverse group of flagellates that are among the dominant protists in aquatic and terrestrial environments (*Boenigk & Arndt, 2002*; *Foissner, 1987*; *Kristiansen & Preisig, 2001*; *Kristiansen & Škaloud, 2017*; *Sandgren, 1988*). These protists serve as excellent models in ecology, ecophysiology, and evolution (*Boenigk, 2008*; *Graupner et al., 2018*), because of their wide range of nutritional strategies. The ecological importance of the chrysophytes is derived from the heterotrophic and mixotrophic taxa being important grazers of bacteria (*Del Campo & Massana, 2011*; *Ekelund, Ronn & Griffiths, 2001*; *Finlay & Esteban, 1998*), and the phototrophic and mixotrophic taxa being a large component of the primary producers in oligotrophic freshwaters (*Kristiansen & Škaloud, 2017*; *Wolfe & Siver, 2013*).

Despite their known ecological importance, chrysophyte taxon richness and species boundaries are difficult to infer. For example, there are some taxa with morphological characters of high diagnostic value such as in *Paraphysomonas* (*Scoble & Cavalier-Smith, 2014*) and *Synura* (*Siver & Lott, 2016*), taxa with morphological characters of uncertain taxonomic value such as in the *Dinobryon divergens* complex (*Jost, Medinger & Boenigk, 2010*), and taxa that are largely missing much morphological characters such as many

Corresponding author
Micah Dunthorn,
micah.dunthorn@uni-due.de

colorless non-scaled taxa (*Grossmann et al., 2016*). Assessing reproductive isolation in these taxa may offer a starting point for a consistent taxonomic revision and recognition of species boundaries based on mating abilities. In general, chrysophytes are assumed to be capable of sex, even though conclusive evidence has not been demonstrated for either meiosis or the fusion of meiotic products from different individuals (*Kristiansen & Škaloud, 2017*). Possible formation of zygotes was observed in *Dinobryon* and *Synura* using morphological observations, but changes in ploidy were not evaluated (*Bourrelly, 1957*; *Fott, 1959*; *Sandgren, 1983*; *Wawrik, 1972*). These morphological studies are also restricted to a handful of taxa and the distribution of sex within the chrysophytes remains unknown.

Meiotic sex is assumed to be retained in most macro-organismic eukaryotes because asexuality can lead to extinction over time (*Bell, 1982*; *Maynard Smith, 1978*). However, sex is often not easily observable in many microbial eukaryotic groups, which can lack distinctive morphological differences between the sexes or we do not know the right environmental conditions to induce sex in the laboratory (*Dunthorn & Katz, 2010*; *Schurko, Neiman & Logsdon, 2009*; *Speijer, Lukeš & Eliáš, 2015*). In the absence of direct observations of sex (*O'Gorman, Fuller & Dyer, 2009*) and in the absence of known sexual mating types (*Corradi & Brachmann, 2017*), one of the strongest molecular signatures of secretive sex in putative asexual protists is the presence of meiotic genes. If the meiotic genes are found in their genomes, then the protein products are likely being used for sex, otherwise they would have been lost over evolutionary time (*Normark, Judson & Moran, 2003*; *Schurko & Logsdon, 2008*). While genomic data are usually used for such meiotic gene inventories in protists (*Chi et al., 2014a*; *Dunthorn et al., 2017*; *Malik et al., 2008*; *Patil et al., 2015*; *Ramesh, Malik & Longsdon, 2005*; *Hofstatter, Brown & Lahr, 2018*), expressed sequence tag (EST) have also been used, although genes can be missing from an EST library if they are not being expressed at the time the protist was collected and analyzed for a secretive sexual stage (*Chi, Parrow & Dunthorn, 2014b*).

Transcriptomic data from 18 chrysophyte isolates, representing 15 different species that were either photo-, mixo-, or hetero-trophic, were recently used to gain insights into nutritional strategies and phylogenetic relationships (*Beisser et al., 2017*). Within the chrysophytes able to perform photosynthesis, the transcriptomes revealed a higher expression of genes participating in photosynthesis, photosynthesis-antenna proteins, porphyrin and chlorophyll metabolism, carbon fixation and carotenoid biosynthesis, while in the heterotrophic strains there was a higher expression of genes involved in nutrient absorption, environmental information processing, and various transporters (e.g., monosaccharide, peptide, and lipid transporters). Here we used those same 18 chrysophyte transcriptomes from *Beisser et al. (2017)* for a meiotic gene inventory to evaluate if these putatively asexual protists are capable of sex. Following *Chi et al. (2014a)*, the presence and absence of these genes were placed into the context that there are two meiotic crossover pathways: class I pathway, which relies on meiotic-specific genes and can include a synaptonemal complex; and class II pathway, which uses meiotic-related genes that are also involved in mitosis (*Loidl, 2016*).

## MATERIALS AND METHODS

From *Beisser et al. (2017)*, sequenced and cleaned transcriptomic data were taken for 18 chrysophytes strains of 15 species: *Acrispumella msimbaziensis* (strain JBAF33), *Apoikiospumella mondseeiensis* (strain JBM08), *Cornospumella fuschlensis* (strain A-R4-D6), *Dinobryon* sp. (strain FU22KAK), *Dinobryon* sp. (strain LO226KS), *Epipyxis* sp. (strain PR26KG), *Ochromonas* or *Spumella* sp. (strain LO244K-D), *Pedospumella encystans* (strain JBMS11), *Poterioochromonas malhamensis* (strain DS), *Poteriospumella lacustris* (strain JBC07), *Poteriospumella lacustris* (strain JBM10), *Poteriospumella lacustris* (strain JBNZ41), *Pedospumella sinomuralis* (strain JBCS23), *Spumella bureschii* (strain JBL14), *Spumella lacusvadosi* (strain JBNZ39), *Spumella vulgaris* (strain 199hm), *Synura* sp. (strain LO234KE), and *Uroglena* sp. (strain WA34KE). The data are available at the European Nucleotide Archive accession PRJEB13662.

Here these data were compared to a query database of nine meiosis-specific and 30 meiosis-related genes established by *Chi et al. (2014a)*. This database was originally established using literature and keyword searches of the NCBI protein database and the Uniprot Knowledgebase. Using local scripts, two methods were used for comparing the transcriptomic data to the query database of meiotic genes: BlastP (*Altschul et al., 1990*) and HMMER v3.0 (*Eddy, 2011*). Reciprocal BLAST analysis was also performed using BLASTP against the non-redundant protein sequence database of NCBI. The parameters for BLASTp and HMMER are default, except sequences were retained if they had hits with *E*-values < 10E-4. Following *Saccharomyces cerevisiae* nomenclature, gene names are signified in italic capital letters, and proteins in lowercase except first letter.

## RESULTS

Out of the 39 meiotic genes, 38 were identified in the transcriptomes of 18 chrysophytes strains (Table 1; File S1). For the nine meiosis-specific genes, all of them were found in at least six transcriptomes. In particular, *SPO11*, which initiates meiosis through double-strand DNA breaks in most eukaryotes (*Keeney, Giroux & Kleckner, 1997*) except in some amoebae (*Bloomfield, 2018*), was found in seven strains. The following other meiosis-specific genes were found: *DMC1* in 15 strains is important for recombination homolog bias (*Bugreev et al., 2011*); *HOP2* in 18 strains stabilizes the association of the protein Dmc1 with DNA (*Chen et al., 2004*); *MND1* in 12 strains also stabilizes the association of the protein Dmc1 with DNA (*Chen et al., 2004*); *HOP1* in six strains, forms part of the synaptonemal complex (*Hollingsworth, Goetsch & Byers, 1990*); *REC8* in 12 strains forms part of the sister chromatid cohesin complex (*Howard-Till et al., 2013*); *MER3* in 16 strains is a DNA helicase (*Nakagawa & Kolodner, 2002*); and *MSH4* in 14 strains and *MSH5* in 13 strains, which are heterodimers that stabilize recombination intermediates (*Nishant et al., 2010*; *Snowden et al., 2004*).

For the 30 meiosis-related genes, 29 were found in at least five out of the 18 transcriptomes. The only gene that was not found in any transcriptome was REC114. The meiosis-related gene *MMS4* was found in the smallest amount of five transcriptomes. The seven meiosis-related genes *MPH1*, *PMS1*, *RAD23*, *RAD50*, *SGS1*, *SMC5*, and

**Table 1 Meiosis genes inventoried in the transcriptomes of 18 strains of 15 species of chrysophytes.**

Chrysophyte species

| Gene | Acrispumella msimbaziensis | Apoikiospumella mondseeiensis | Cornospumella fuschlensis | Dinobryon sp. (strain FU22KAK) | Dinobryon sp. (strain LO226KS) | Epipyxis sp. | Ochromonas or Spumella sp. | Pedospumella encystans | Pedospumella sinomuralis | Poteriochromonas malhemensis | Poteriospumella lacustris (strain JB07) | Poteriospumella lacustris (strain JBM10) | Poteriospumella lacustris (strain JBNZ41) | Spumella bureschii | Spumella lacusvadosi | Spumella vulgaris | Synura sp. | Uroglena sp. |
|---|---|---|---|---|---|---|---|---|---|---|---|---|---|---|---|---|---|---|
| **Double-strand break formation** | | | | | | | | | | | | | | | | | | |
| REC114 | − | − | − | − | − | − | − | − | − | − | − | − | − | − | − | − | − | − |
| SPO11 | − | − | + | − | − | + | − | + | + | − | − | − | − | + | − | + | + | − |
| **Crossover regulation** | | | | | | | | | | | | | | | | | | |
| DMC1 | + | + | + | + | − | + | + | + | + | + | + | + | + | + | + | + | − | − |
| HOP1 | − | − | − | − | − | + | − | + | − | − | + | + | − | + | − | + | − | + |
| HOP2 | + | + | + | + | + | + | + | + | + | + | + | + | + | + | + | + | + | + |
| MER3 | − | + | + | + | − | + | + | + | + | + | + | + | + | + | + | + | + | + |
| MND1 | − | − | + | + | − | + | − | + | − | + | + | + | + | + | + | − | + | + |
| MSH4 | − | + | + | + | − | + | − | + | + | + | + | + | + | + | + | + | + | − |
| MSH5 | − | − | + | + | − | + | − | + | + | + | + | + | + | + | + | + | + | − |
| **Double-strand break repair** | | | | | | | | | | | | | | | | | | |
| REC8 | − | − | + | + | − | + | − | + | + | + | + | + | + | + | + | + | − | − |
| **Bouquet formation** | | | | | | | | | | | | | | | | | | |
| SAD1 | − | − | + | + | − | + | − | + | + | + | + | + | + | + | + | + | − | + |
| **DNA damage sensing/response** | | | | | | | | | | | | | | | | | | |
| MRE11 | − | − | + | + | − | + | − | + | + | + | + | + | − | + | − | + | − | − |
| RAD17 | + | + | + | + | − | + | + | + | + | + | + | + | + | + | + | − | + | − |
| RAD23 | + | + | + | + | + | + | + | + | + | + | + | + | + | + | + | + | + | + |
| RAD24 | + | − | + | + | + | + | + | + | + | + | + | + | + | + | + | + | + | + |
| RAD50 | + | + | + | + | + | + | + | + | + | + | + | + | + | + | + | + | + | + |
| NBS1 | − | − | + | + | − | − | − | + | + | + | + | + | + | + | − | − | − | + |
| **Double-strand break repair (nonhomology end join)** | | | | | | | | | | | | | | | | | | |
| KU | − | − | + | + | − | + | − | + | + | + | + | + | + | − | + | + | + | − |
| LIG4 | + | − | + | + | − | + | − | + | + | + | + | + | + | + | + | + | − | + |
| LIF1 | − | − | − | − | − | − | − | − | − | − | − | − | − | − | − | − | − | − |
| **Recombinational repair** | | | | | | | | | | | | | | | | | | |
| DNA2 | − | + | + | + | − | + | − | + | + | + | + | + | + | + | + | + | + | + |
| MMS4 | − | − | − | − | − | + | − | + | − | − | − | − | − | + | − | + | + | − |

| Gene | Acrispumella msimbaziensis | Apoikiospumella mondseeiensis | Cornospumella fuschlensis | Dinobryon sp. (strain FU22KAK) | Dinobryon sp. (strain LO226KS) | Epipyxis sp. | Ochromonas or Spumella sp. | Pedospumella encystans | Pedospumella sinomuralis | Poteriochromonas malhemensis | Poteriospumella lacustris (strain JB07) | Poteriospumella lacustris (strain JBM10) | Poteriospumella lacustris (strain JBNZ41) | Spumella bureschii | Spumella lacusvadosi | Spumella vulgaris | Synura sp. | Uroglena sp. |
|---|---|---|---|---|---|---|---|---|---|---|---|---|---|---|---|---|---|---|
| EME1 | – | – | – | – | – | – | – | – | – | – | – | – | – | – | – | – | – | – |
| EXO1 | – | – | + | + | – | + | – | + | + | + | + | + | + | + | – | + | + | + |
| FEN1 | + | – | + | + | + | + | + | + | + | + | + | + | + | + | + | + | + | + |
| MLH1 | + | – | + | + | – | – | + | + | + | + | + | – | + | + | + | + | + | – |
| MLH3 | – | + | + | + | + | + | + | + | + | + | + | + | + | + | + | + | + | – |
| MPH1 | + | + | + | + | + | + | + | + | + | + | + | + | + | + | + | + | + | + |
| MSH2 | – | – | + | + | – | + | – | + | + | + | + | + | + | + | + | + | + | – |
| MSH6 | + | – | + | + | – | + | – | + | + | + | + | + | + | + | + | + | + | + |
| MUS81 | – | – | + | + | – | + | + | + | + | + | + | + | + | + | + | + | – | – |
| PMS1 | + | + | + | + | + | + | + | + | + | + | + | + | + | + | + | + | + | + |
| RAD51 | + | + | + | + | – | + | + | + | + | + | + | + | + | + | + | – | – | – |
| RAD52 | – | – | – | + | – | – | – | – | – | – | + | + | – | + | – | – | – | – |
| RAD54 | – | + | + | + | + | + | + | + | + | + | + | + | + | + | + | + | + | + |
| RTEL | – | – | + | + | – | + | – | + | + | + | + | + | + | + | – | + | – | – |
| SAE2 | + | – | + | + | – | + | + | + | + | + | + | + | + | + | + | + | + | + |
| SLX1 | – | – | + | + | – | + | – | + | + | – | + | + | + | + | – | – | + | – |
| SLX4 | + | – | + | – | + | + | + | + | + | – | – | – | – | + | + | + | – | – |
| SMC5 | + | + | + | + | + | + | + | + | + | + | + | + | + | + | + | + | + | + |
| SMC6 | + | + | + | + | + | + | + | + | + | + | + | + | + | + | + | + | + | + |
| GEN1 | – | – | + | + | – | + | – | + | + | + | + | + | + | + | – | + | + | – |

SMC6 were found in all 18 transcriptomes. Nine of the other meiosis-related genes were only not present in two or three chrysophyte transcriptomes.

Many of the missing meiotic genes could really be missing from the genomes, or the genes could be missing because of how the data were generated. In transcriptomes, just like in ESTs (*Chi, Parrow & Dunthorn, 2014b*), missing genes are expected because only genes being actively expressed will be sequenced. These differences between the sequences of strains of the same species here suggest that indeed the transcriptomes are likely missing a lot of non-expressed genes. For example, *HOP1* is only found in two of three strains of *Poteriospumella lacustris*, and *MSH4* and *MSH5* are only found in one of two stains of *Dinobryon* sp.

## DISCUSSION

In this gene inventory of chrysophyte transcriptomes, we found evidence for the presence of many meiosis-specific and meiosis-related genes. If we assume a use-it-or-lose-it view of these genes (*Normark, Judson & Moran, 2003*; *Schurko & Logsdon, 2008*), then the chrysophytes are using the protein products of these genes to construct functional meiotic machinery. As with most other eukaryotes (*Dunthorn & Katz, 2010*; *O'Malley, Simpson & Roger, 2013*), the chrysophytes are therefore likely sexual, which supports earlier microscopic observations that potentially indicated sex (*Bourrelly, 1957*; *Fott, 1959*; *Sandgren, 1983*; *Wawrik, 1972*). If this is the case, and even if sex has not yet been directly observed, the genetic diversity and adaptive evolution of the chrysophytes would benefit from this secretive sex. And this benefit could occur even if sex was a rare event in the chrysophytes (*D'Souza & Michiels, 2010*; *Green & Noakes, 1995*).

Additionally, we found meiotic genes involved in both crossover pathways, including genes involved in making the synaptonemal complex in class I pathway. Although these pathways have been differentially lost in various eukaryotic groups (*Chi et al., 2014a*; *Loidl, 2016*), chrysophyte potentially use both of these pathways. Given the phylogenetic placement across the chrysophyte tree of life of the 15 species sampled here (*Beisser et al., 2017*), these results supporting secretive sex and the presence of both crossover pathways should be applicable for all, or most, other chrysophyte species.

Here we used transcriptomic data to show that there are meiotic genes in the putative asexual chrysophytes. These genes are likely being used for sex. This finding suggests that more thorough *de novo* genome sequencing of different chrysophyte species should be performed to uncover the meiotic genes possibly missed in the transcriptomes. This finding also suggests that targeted mating attempts of different chrysophyte species in the laboratory should be attempted, as these observations will offer the best evidence that the chrysophytes are truly sexual in nature and that meiosis in these protists is not being used just for automixis.

### Funding

The Deutsche Forschungsgemeinschaft provided support to Micah Dunthorn (grant #DU1319/1-1) and Jens Boenigk (grant #s BO3245/17 and BO3245/19). The funders had no role in study design, data collection and analysis, decision to publish, or preparation of the manuscript.

### Grant Disclosure

The following grant information was disclosed by the authors:
Deutsche Forschungsgemeinschaft: DU1319/1-1, BO3245/17, and BO3245/19.

### Competing Interests

The authors declare that they have no competing interests.
_______________________

## Author Contributions

- Diana Kraus conceived and designed the experiments, performed the experiments, analyzed the data, prepared figures and/or tables, authored or reviewed drafts of the paper, approved the final draft.
- Jingyun Chi performed the experiments, analyzed the data, contributed reagents/materials/analysis tools, prepared figures and/or tables, authored or reviewed drafts of the paper, approved the final draft.
- Jens Boenigk analyzed the data, contributed reagents/materials/analysis tools, authored or reviewed drafts of the paper, approved the final draft.
- Daniela Beisser analyzed the data, contributed reagents/materials/analysis tools, approved the final draft.
- Nadine Graupner analyzed the data, contributed reagents/materials/analysis tools, approved the final draft.
- Micah Dunthorn conceived and designed the experiments, analyzed the data, prepared figures and/or tables, authored or reviewed drafts of the paper.

## Data Availability

European Nucleotide Archive accession PRJEB13662.

## Supplemental Information

Supplemental information for this article can be found online at http://dx.doi.org/10.7717/peerj.5894#supplemental-information.

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
