# Peer review of "Putatively asexual chrysophytes have meiotic genes: evidence from transcriptomic data"

_PeerJ, doi:10.7717/peerj.5894_

## Round 0.1 · original submission · Minor Revisions

Dear Dr. Kraus and colleagues:

Thanks for submitting your manuscript to PeerJ. I have now received two independent reviews of your work, and as you will see, both are favorable. Well done! Nonetheless, both reviewers raised some minor concerns about the research, and areas where the manuscript can be improved. I agree with the reviewers, and thus feel that their concerns should be adequately addressed before moving forward.

Therefore, I am recommending that you revise your manuscript accordingly, taking into account all of the issues raised by the reviewers. I do believe that your manuscript will be ready for publication once these issues are addressed.

Good luck with your revision,

-joe

Reviewer 1 ·

Basic reporting

no comment

Experimental design

Good

Validity of the findings

no comment

Additional comments

Putatively asexual chrysophytes have meiotic genes: evidence from transcriptomic data

The authors present a short report on inventory of meiosis and sex related genes from 15 representatives of chrysophytes using transcriptome data. Their findings show that chrysophytes are expressing sex genes and hence are likely sexual lineages through some obscure mechanism.

This report is a good addition in supporting the ancestral nature of sex in eukaryotes. While the goals of this manuscript are clear there some inconsistencies and sections that need clarification.

Major and minor comments are list below for consideration for the authors.

1. The introduction clearly states there is plenty of literature indicating that chrysophytes are sexual lineages (lines 47-53), Yet the authors claim that it is putatively asexual including in the title. This is confusing. Their results also clearly demonstrate chrysophytes are sexual lineages (almost all inventoried genes have been detected). What is new here besides the lack of knowledge on the mechanism of sexual development? If the authors only analyzed lineages that are reportedly asexual they need to clarify this and give some of kind background information on those lineages.
2. The references in this article is limited to self citation it would be nice if they can expand their references to include some recent gene inventory works in other microbial eukaryotes that used similar datasets and approach.
3. Materials and methods – the data used in this study was from previously published work. It is not clear why the authors included methods on RNA collection and sequencing.
4. Lines 149-151: ‘WITH’ missing or revise sentence
“Although these150 pathways have been differentially lost in various eukaryotic groups (Chi et al. 2014a; Loidl 2016), chrysophyte potentially can go down WITH both of them.”
5. Last sentence not clear how the findings of this study could possibly help in this aspect. Please elaborate or remove
‘This finding can also be used to targetmating attempts of different chrysophyte species in the laboratory, as these observations will offer the best evidence that the chrysophyte are truly sexual in nature.’

Reviewer 2 ·

Basic reporting

The paper by Kraus et al. („Putatively asexual chrysophytes have meiotic genes: evidence from transcriptomic data“) investigated the expression of nine meiosis-specific and 30 meiosis-related genes in chrysophytes - eukaryotic, single-celled organisms. They matched transcriptomic data with genes from an already published databank used for similar analyses on other protists (Cilliata and Dinoflagellata). Transcriptomic data of the 15 chrysophyte species used were also published earlier. So this is a nice study combining existing data to investigate a specific trait. The benefit of the used method is that this trait can be quickly assessed across a range of related taxa, without difficult or impossible morphological observations.

The paper is clearly written and well structured. The introduction and background are easy to read.
The authors explain what they use as indication for sexual reproduction in crysophytes (i.e. cyst or zygote formation) which has only been observed in a few taxa.
Next to sexual reproduction, it would be interesting to understand the asexual reproduction of these organisms. Sometimes asexual reproduction also requires meiosis and meiotic genes remain functional despite being asexual (e.g. autmixis). Is it possible that the author exclude this possibility, i.e. the use of meiotic genes in asxual reproduction, in their introduction?

Experimental design

The paper generated the following data:
1) local script comparing transcriptomes against a meiotic-genes database
2) fasta sequences representing positive Blast hits, including e-values, for each strain/species

The material and method section is short and precise, though it would be helpful for reproducability of the methods, if:

a) authors could provide parameters they used for their BLAST and HMMER search. If default settings were used, it would also be nice to know.

b) which e-value threshold (similarity threshold) they used to evaluate if genes from transcriptomes were actually functional and orthologue to genes from the query database. I assume similarity of genes varied among species and were not identical to the database. E-values were provided with the fasta files in the supplementary files, but without explanation the reader is not able to understand if the e-value represents a good, mediocre or by chance hit.

c) making the local scripts available in the supplementary information would be beneficial for the manuscript

d) to know if any paralogues were detected

The experimental design is sound and was adapted from earlier publications on similar topics.
The research question is also well-defined and is definitely relevant. Shortcomings of this methodology and dataset are openly discussed.
Methods would benefit from more transparency as mention in the section above.

Validity of the findings

Impact and novelty of this approach are limited but the general idea of this research is beneficial for the scientific community working with protists and the field of sexual reproduction, in particular in evolutionary contexts.
As far as I could see as a reviewer the data are robust and sound, but methods are too short to judge eventually.
Discussion and Conclusions are sound and well-written.

---

## Round 0.2 · accepted · Accept

Dear Dr. Kraus and colleagues:

Thanks for re-submitting your manuscript to PeerJ, and for addressing the concerns raised by the reviewers. I now believe that your manuscript is suitable for publication. Congratulations! I look forward to seeing this work in print, and I anticipate it being an important resource for the chrysophyte community, as well as researchers with interests in the mechanisms underpinning asexual reproduction. Thanks again for choosing PeerJ to publish such important work.

Best,

-joe

#